# Exact Shape Correspondence via 2D Graph Convolution

**Barakeel Fanseu Kamhoua**[1], **Lin Zhang**[2*], **Yongqiang Chen**[1], **Han Yang**[1], **Kaili Ma**[1],
**Bo Han**[3], **Bo Li**[2], **James Cheng**[1]
[1]The Chinese University of Hong Kong, [2]The Hong Kong University of Science and Technology,
[3]Hong Kong Baptist University

## Abstract

For exact 3D shape correspondence (matching or alignment), i.e., the task of matching each point on a shape to its exact corresponding point on the other shape (or to be more specific, matching at geodesic error 0), most existing methods do not perform well due to two main problems. First, on nearly-isometric shapes (i.e., low noise levels), most existing methods use the eigen-vectors (eigen-functions) of the Laplace Beltrami Operator (LBO) or other shape descriptors to update an initialized correspondence which is not exact, leading to an accumulation of update errors. Thus, though the final correspondence may generally be smooth, it is generally inexact. Second, on non-isometric shapes (noisy shapes), existing methods are generally not robust to noise as they usually assume near-isometry. In addition, existing methods that attempt to address the non-isometric shape problem (e.g., GRAMPA) are generally computationally expensive and do not generalise to nearly-isometric shapes. To address these two problems, we propose a 2D graph convolution-based framework called 2D-GEM. 2D-GEM is robust to noise on non-isometric shapes and with a few additional constraints, it also addresses the errors in the update on nearly-isometric shapes. We demonstrate the effectiveness of 2D-GEM by achieving a high accuracy of 90.5% at geodesic error 0 on the non-isometric benchmark SHREC16, i.e., TOPKIDS (while being much faster than GRAMPA), and on nearly-isometric benchmarks by achieving a high accuracy of 92.5% on TOSCA and 84.9% on SCAPE at geodesic error 0.

## 1 Introduction

3D shapes can undergo different types of deformations. Deformations can be rigid or non-rigid. Rigid deformations include rotations, translations, reflections, or any composition of these resulting in pose changes. They preserve geometric properties of the shape such as segment lengths, angles, volumes, sizes, geodesic distances and others. Non-rigid deformations do not preserve geometric properties of the shape, they include dilations, or a composition of dilations and rigid deformations (e.g., stretching, scaling). In this work we focus on non-rigid shapes that are either nearly-isometric (i.e., most geometric properties are preserved), or non-isometric (i.e., geometric properties are not preserved, and contain strong topological noise).

Non-rigid shape matching is a very important task in machine learning, shape analysis and computer vision. It has important applications such as shape registration, comparison, recognition, and retrieval. 3D shapes are usually represented as 2D manifolds embedded in $\mathbb{R}^3$. Given two three dimensional shapes $\mathcal{M}_1$ and $\mathcal{M}_2$ with $n$ vertices each, the goal of non-rigid shape matching is to find a meaningful correspondence $\mathcal{T} : \mathcal{M}_1 \to \mathcal{M}_2$, where $\mathcal{T}$ is (a) bijective, (b) continuous in both directions in the sense that nearby points on $\mathcal{M}_1$ should be matched to nearby points on $\mathcal{M}_2$ (vice versa), and (c)

---

*Corresponding author: lzhangbv@connect.ust.hk

36th Conference on Neural Information Processing Systems (NeurIPS 2022).

similar points should be matched to each other [42]. In the discrete setting the correspondence (map or match) $\mathcal{T}$ can be represented as a permutation matrix $\mathbf{P} \in \{0, 1\}^{n \times n}$ [54, 4].

In more details, 3D shapes are usually represented using triangulations (e.g., the Delauney triangulation [20]) and xyz-coordinates representations in $\mathbb{R}^3$ [80, 42]. It is hard to utilize these xyz-coordinates and triangulations to directly perform 3D shape matching because of: (a) embedding ambiguities, i.e. shapes sharing the same metric (isometry) or very similar metrics (nearly-isometric) can have drastically different xyz-coordinates representations in $\mathbb{R}^3$ due to pose changes [80], and (b) non-isometric shapes with strong topological noise will have even more different xyz-coordinates and triangulations.

As such, these 3D-coordinates together with the triangulations are usually used to extract *point-wise descriptors* (which encode the similarity between points), and *pair-wise descriptors* (which encode global and/or local relations between pairs of points). Once these descriptors are built, the shape matching problem is reduced to an alignment problem over (a) the *point-wise descriptors*, or (b) the *pair-wise descriptors*, or (c) *both (joint alignment)*. The *joint alignment* problem can be written as:

$$\mathbf{P} = \arg \min_{\mathbf{P} \in \Pi} \mathbb{E}(\mathbf{P}) = \arg \min_{\mathbf{P} \in \Pi} \mathbb{F}(\mathbf{P}) + \alpha \mathbb{H}(\mathbf{P}), \tag{1}$$

where $\Pi$ is the set of permutation matrices, $\mathbb{E}$ is the generalised energy, $\mathbb{F}$ is the energy over *point-wise descriptors*, $\mathbb{H}$ is the energy over *pair-wise descriptors*, and $\alpha$ is a trade-off parameter [42]. Several algorithms have been proposed to address point-wise alignment, [4, 70, 11, 55, 64], pair-wise alignment [20, 18], as well as joint alignment [59, 57, 56, 2, 79, 42, 54, 58]. Due to space constraints, see Appendix A for more details on the different alignments as well as related works.

To measure the performance of an algorithm on the 3D shape matching task, the geodesic error is usually used as the error metric [20, 80]. Given that a point $x$ on shape $\mathcal{M}_1$ is mapped by an algorithm to $y$ as its correspondence on shape $\mathcal{M}_2$, the geodesic error measures how far the match $y$ is from the true matching position for $x$ on $\mathcal{M}_2$ i.e., $y^*$ (given by the true map). When the map returned by algorithm is exact, it significantly improves performance and efficiency of other tasks such as registration, comparison, motion transfer, style transfer. However, most methods proposed in each of the aforementioned categories for 3D shape matching do not perform well at geodesic error 0 (i.e., the task of exact matching where $y$ is the same point $y^*$). This is mainly due to two factors.

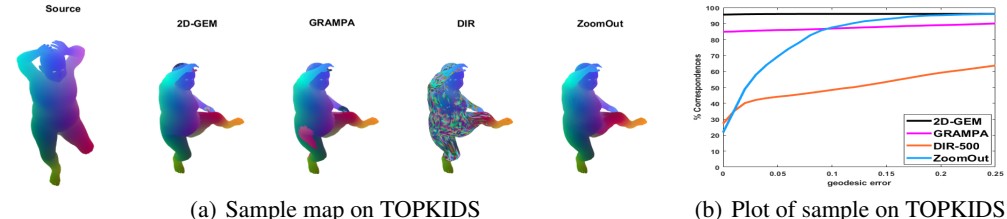

(a) Sample map on TOPKIDS      (b) Plot of sample on TOPKIDS

Figure 1: A non-isometric example. Illustrating the performance of 2D-GEM and other methods on a sample shape from the TOPKIDS dataset. For DIR we used DIR-500 with all LMD thresholds set to 5 since geodesic distances are not conserved on TOPKIDS (note that here, DIR performs much worse with lower thresholds). For ZoomOut we used 200 eigen-values and started from 20 with a step size of 1. For 2D-GEM and GRAMPA we followed the parameters in Section D.4

First, on non-isometric shapes, given that most existing methods assume near-isometry, they cannot handle noisy shapes (e.g., shapes with strong topological noise such as mesh "gluing" in areas of contact as in the TOPKIDS dataset [42]), since in this case geometric properties such as geodesic distances are not preserved and hence the shapes are not nearly isometric. An illustration can be seen in Figures 1(a) and 1(b), where the existing methods that assume near-isometry such as ZoomOut [54] and DIR [80] do not perform well at geodesic error 0 on the sample shape from TOPKIDS. They even produce a non-smooth map in the case of DIR. Moreover, those methods that try to address this noise are computationally very expensive and do not generalize well to nearly-isometric shapes, because for nearly-isometric shapes stronger constraints (such as preserving geodesic distances) are needed for the match. To illustrate this point, the spectral method GRAMPA is very expensive as it needs to compute all the eigen-values and eigen-vectors of the adjacency matrices obtained from the triangulations of each shape (see more details in Appendix B and time comparison with 2D-GEM in Figures 2(a) and 2(b)). Moreover, GRAMPA cannot generalize to cases of near-isometry

as illustrated in Figure 3(a), which shows the matching of GRAMPA in an example from SCAPE, a nearly-isometric shape dataset).

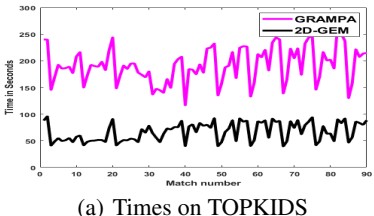

(a) Times on TOPKIDS

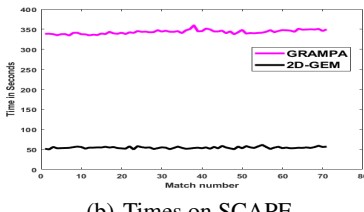

(b) Times on SCAPE

Figure 2: Time taken by 2D-GEM and GRAMPA for each of the 90 shapes on TOPKIDS and 71 on SCAPE.

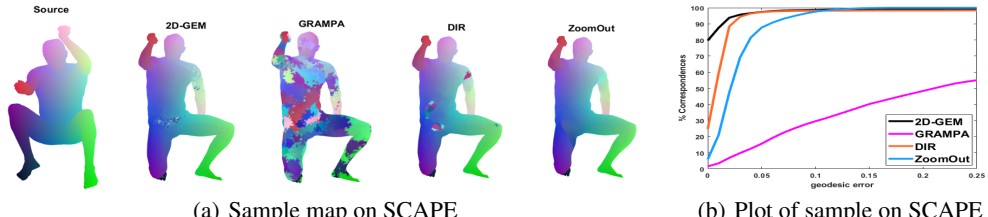

(a) Sample map on SCAPE    (b) Plot of sample on SCAPE

Figure 3: An isometric example. Illustrating the performance of 2D-GEM and other methods on a sample shape from the SCAPE dataset (using same settings as in figure 1).

Second, for nearly-isometric shapes, most existing methods attempt to first find an initial correspondence and then iteratively improve this initial correspondence using all points and the eigen-vectors corresponding to lower frequencies (eigen-values) of the LBO [4, 58, 54] or using other shape descriptors [64]. However, such an approach leads to an accumulation of errors in the correspondence, since the initial correspondence is mostly not accurate. This problem has been discussed in details by Xiang et al. [80], which attempted to address this via the Dual Iterative Refinement (DIR). DIR finds well matched points using the notion of Local Mapping Distortion (LMD) to identify points for which the geodesic distances are well preserved by the correspondence, and then use these points to update the correspondences of the other points via the notion of Functional maps [56]. Thus, DIR reduces the error accumulation. However, the DIR approach only works for nearly-isometric shapes and not for non-isometric shapes since it is based on the assumption of preserving geodesic distances which are not preserved in the case of non-isometries (see Figure 1(a), which shows that DIR does not perform well at geodesic 0 for non-isometric shapes). Finally, even on nearly-isometric shapes, DIR, though effective, requires a lot of eigen-vectors of the LBO in order to perform well at geodesic error 0. This can be seen in Figure 3(b), where using up to 500 eigen-vectors gives only $25\%$ accuracy at geodesic error 0 on the sample shape from SCAPE, and even with up to 1000 eigen-vectors it only achieves $69.5\%$ correspondence accuracy on SCAPE and a $59.8\%$ on TOSCA at geodesic error 0 (as reported in the experiments in Section 5).

Given these problems, we propose to utilize 2D graph convolution for (a) non-isometric shape matching (since it is a de-noising operation), and (b) for isometric shape matching by further constraining our framework to preserve geodesic distances.

To this end:

- we show how graph matching can be done via the 2D graph convolution, and re-define noise in the context of graph matching. We further show empirically that this approach can match the recent spectral method GRAMPA on Erdos Regny random graph matching, as well as outperform it in the presence of informative node features (see Appendix D),

- we propose a new 2D graph convolution-based approach to shape matching, called 2D-GEM, which is (a) robust to noise, and achieves high accuracy on exact matching of non-isometric shapes while being orders of magnitude faster than GRAMPA, and (b) with some additional constraints also achieves high accuracy on exact matching of nearly-isometric shapes.

## 2 2D graph Convolution for Joint Alignment

To the best of our knowledge, we are the first to propose a 2-D graph convolution for graph alignment and shape matching. The most closely related work is the work that uses graph convolution neural networks for graph alignment [22] (See Appendix A.2) Here, we revisit the traditional graph convolution, show how the *joint alignment* problem can be done via 2D graph convolution, and finally we re-define noise in the context of graph matching.

### 2.1 Graph Convolution

From the Graph Signal Processing [67] perspective, graph convolution is regarded as a frequency filtering operator over node attributes. Given a graph, with node attribute matrix $\mathbf{X} \in \mathbb{R}^{n \times d}$, its normalized symmetric Laplacian (referred to as Laplacian in the rest of the paper) can be eigen-decomposed as $\mathbf{L} = \mathbf{U}\Lambda\mathbf{U}^T$, where $\Lambda = \text{diag}(\lambda_1, \cdots, \lambda_n) \in \mathbb{R}^{n \times n}$ are the eigen-values in increasing order, and $\mathbf{U} = [\mathbf{u}_1, \cdots, \mathbf{u}_n] \in \mathbb{R}^{n \times n}$ are the associated orthogonal eigen-vectors. The graph convolution operation applied to the graph signal $\mathbf{x}$ (a column of $\mathbf{X}$) consists of three steps [45, 24, 29, 30]: (1) transform node attributes into the spectral domain (i.e., $\mathbf{z} = \mathbf{U}^T\mathbf{x}$); (2) remove attribute noise by scaling the spectrum with a function defined on eigen-values (i.e., $\bar{z}_i = g(\lambda_i)z_i$); (3) perform inverse transformation on the scaled spectrum (i.e., $\bar{\mathbf{x}} = \mathbf{U}\bar{\mathbf{z}}$), where $g(\lambda)$ is a scalar-valued function on the eigen-values $\lambda$. In short, graph convolution is able to remove the noisy signals by assigning smaller weight on higher eigen-value spectrum with a well defined function $g(\lambda)$. For convenience, we denote $g(\mathbf{L}) = \mathbf{U}g(\Lambda)\mathbf{U}^T$, so that we have the convolution as $g * \mathbf{X} = g(\mathbf{L})\mathbf{X}$. We refer to $g(\mathbf{L})$ as the *convolution function* and $g(\Lambda)$ as the *filter function* or simply filter.

### 2.2 Joint Alignment via 2D graph Convolution

Following previous works [22, 71] and Appendix C, the graph convolution is invariant to permutation. As such, given two graphs $\mathcal{A}$ and $\mathcal{B}$ with graph convolution functions $\mathbb{M}_A = g(\mathbf{L}_A), \mathbb{M}_B = h(\mathbf{L}_B) \in \mathbb{R}^{n \times n}$, features $\mathbf{X}_A, \mathbf{X}_B \in \mathbb{R}^{n \times d}$, and having the functions $g$ and $h$ to be the same function, the *joint alignment* problem in Equation (1) can instead be performed as:

$$\min_{\mathbf{P}} ||\mathbf{P}\mathbb{M}_A\mathbf{X}_A - \mathbb{M}_B\mathbf{X}_B||_F^2, \tag{2}$$

see Appendix C for more details. We observe that (2) reduces to the Linear Assignment Problem (LAP):

$$\max_{\mathbf{P}} \mathbf{Tr}(\mathbf{P}\mathbb{M}_A\mathbf{X}_A\mathbf{X}_B^T\mathbb{M}_B), \tag{3}$$

where $\mathbf{Tr}(\cdot)$ is the matrix trace operator, the convolution functions capture the pair-wise descriptors, and the features capture the point-wise descriptors. The first benefit of the convolution is to help do the joint alignment as a Linear Assignment Problem (LAP), which is much cheaper than the traditional Quadratic Assignment Problem (QAP) [23] used for graph matching. Note that, $\mathbb{M}_A = \mathbf{U}g(\Lambda)\mathbf{U}^T = \sum_i g(\lambda_i)\mathbf{u}_i\mathbf{u}_i^T$ and $\mathbb{M}_B = \mathbf{V}h(\mathbf{M})\mathbf{V}^T = \sum_j h(\mu_j)\mathbf{v}_j\mathbf{v}_j^T$, where $\mathbf{U}, \mathbf{V}$ are the orthogonal eigen-vectors of $\mathbb{M}_A$ and $\mathbb{M}_B$, and $\Lambda, \mathbf{M}$ are the diagonal matrices of their respective eigen-values. Given these, one can rewrite our cost matrix $\mathbf{C} = \mathbb{M}_A\mathbf{X}_A\mathbf{X}_B^T\mathbb{M}_B$ in (3) as:

$$\mathbf{C} = \sum_{i,j} g(\lambda_i)h(\mu_j)\mathbf{u}_i\mathbf{u}_i^T\mathbf{X}_A\mathbf{X}_B^T\mathbf{v}_j\mathbf{v}_j^T, \tag{4}$$

When node features are not available, one can initialize all nodes with the same features in (3), i.e., $\mathbf{X}_A = \mathbf{J}_A \in \mathbb{R}^{n_1 \times d}$, and $\mathbf{X}_B = \mathbf{J}_B \in \mathbb{R}^{n_2 \times d}$, where $n_1$ may or may not be equal to $n_2$ for different size graphs, and $\mathbf{J}_A, \mathbf{J}_B$ are all one rectangular matrices. This is equivalent to assuming that nodes have the same probability of being matched to each other. Moreover, to save cost in the case when there is no feature, one could simply initialize the similarity matrix $\mathbf{X}_A\mathbf{X}_B^T = \mathbf{J}_{A,B} \in \mathbb{R}^{n_1 \times n_2}$ (given that $\mathbf{J}_A\mathbf{J}_B^T = \mathbf{J}_{A,B} \in \mathbb{R}^{n_1 \times n_2}$, where $\mathbf{J}_{A,B}$ is a scalar matrix with constant $[\mathbf{J}_{A,B}]_{ij} = \sum_{i=1}^d 1 = d$) for all entries. In the case where there is no feature, (4) becomes:

$$\mathbf{C} = \sum_{i,j} g(\lambda_i)h(\mu_j)\mathbf{u}_i\mathbf{u}_i^T\mathbf{J}_{A,B}\mathbf{v}_j\mathbf{v}_j^T. \tag{5}$$

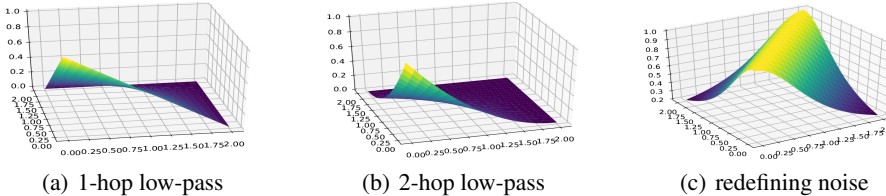

|  (a) 1-hop low-pass | (b) 2-hop low-pass | (c) redefining noise |

Figure 4: 2D graph filters. Figure 4(a) uses the convolution functions $g(\mathbf{L}_A) = g(\mathbf{L}_B) = (\mathbf{I} - \frac{1}{2}\mathbf{L})$, Figure 4(b) uses the convolution functions $g(\mathbf{L}_A) = g(\mathbf{L}_B) = (\mathbf{I} - \frac{1}{2}\mathbf{L})^2$, and Figure 4(c) uses the noise robust filter defined in Section 3 with $e = 1$.

Our cost matrices in equations (4) and (5) can be seen as 2D graph convolution on the 2D graph signals $\mathbf{X}_A\mathbf{X}_B^T$ (or $\mathbf{J}_{A,B}$ in the case of no features) [46], where we: (1) transform the 2D graph signals into the spectral domain to obtain the 2D graph spectrum (i.e., $\mathbf{Z} = \mathbf{U}^T\mathbf{X}_A\mathbf{X}_B^T\mathbf{V}$ or $\mathbf{Z} = \mathbf{U}^T\mathbf{J}_{A,B}^T\mathbf{V}$) [46]; (2) remove noise (de-noise) from the 2D graph signals by scaling the 2D spectrum with a function defined on each pair of eigen-values (i.e., $\bar{z}_{ij} = w(\lambda_i, \mu_j)z_{ij}$); and (3) perform inverse transformation on the scaled 2D spectrum ($\mathbf{C} = \mathbf{U}\bar{\mathbf{Z}}\mathbf{V}^T$). Thus, our cost matrices are 2D filtering operation on the 2D graph signals, where $w(\lambda_i, \mu_j) = g(\lambda_i)h(\mu_j)$ is the 2D filter (see Figures 4(a), 4(b) and 4(c) for different 2D filters).

## 3 Redefining Noise in the Context of Graph Matching

In this section, we redefine noise in the context of graph matching and propose a suitable filter for graph convolution with regards to the graph matching task . It can be shown that graph convolution is a solution to the graph signal de-noising problem [24]. Formally, assuming that (1) the ground-truth node features $\hat{\mathbf{X}}$ are smooth w.r.t. a graph $\mathcal{G}$ with adjacency $\mathbf{A}$ and Laplacian $\mathbf{L}$, and that (2) the node feature noise has small magnitude, then the total variation of $\hat{\mathbf{X}}$ should be small by assumption (1), and $||\hat{\mathbf{X}} - \mathbf{X}||$ can be upper-bounded by assumption (2). Thus, we can model the problem of recovering the ground-truth signal as:

$$\hat{\mathbf{X}}^* = \arg\min_{\hat{\mathbf{X}}} \mathbf{Tr}\left(\hat{\mathbf{X}}^\top\mathbf{L}\hat{\mathbf{X}}\right) \qquad \text{s.t.} ||\hat{\mathbf{X}} - \mathbf{X}||_F^2 \leq \epsilon_1, \qquad (6)$$

It can clearly be observed that $\hat{\mathbf{X}}^* = (\mathbf{I} + \alpha\mathbf{L})^{-1}\mathbf{X}$, where $\alpha$ is a hyperparameter. This is a graph convolution with graph convolution function given by $(\mathbf{I} + \alpha\mathbf{L})^{-1}$. This function can be decomposed as $\mathbf{U}(\mathbf{I} + \alpha\Lambda)^{-1}\mathbf{U}^T$. As well studied in [45, 78, 36], one can see that the filter $(\mathbf{I} + \alpha\Lambda)^{-1}$ is decaying, i.e., $g(\lambda_i)$ is lower for higher $\lambda_i$, which will mostly reduce the amplitude $z_i$ of graph signals $\mathbf{u}_i$ belonging to larger eigen-values $\lambda_i$. Thus, one can conclude that noise corresponds to higher order frequencies (eigen-values). This kind of filter is commonly known as a low-pass filter [45]. It can be shown that for the 2D graph convolution, filters of the form $w(\lambda_i, \mu_j) = g(\lambda_i)h(\mu_i)$, where $g(\lambda_i) = (1 - \alpha\lambda_i)^k$ and $h(\mu_j) = (1 - \alpha\mu_j)^k$ will result in a 2D low-pass filter as well (see Figures 4(a) and 4(b)). These filters are referred to as $k$-hop low-pass filters (aggregating information from the $k$-hop neighborhood of a node).

However, in the context of graph matching, graphs are isomorphic when they have the same eigen-value, and in this case we want to align the eigen-vectors since two graphs $\mathcal{A}$ and $\mathcal{B}$ are isomorphic iff their eigen-vectors are permutations of each other up to a sign [72, 37].

This leads us to redefine noise in the graph matching context. Given two graphs $\mathcal{A}$ and $\mathcal{B}$, with convolution functions $\mathbb{M}_A$, $\mathbb{M}_B$ whose eigen-values are $\lambda_i$ and $\mu_j$ respectively, noise in graph matching corresponds to 2D graph signals $z_{i,j} = \mathbf{u}_i^T\mathbf{X}_A\mathbf{X}_B^T\mathbf{v}_j$ or $z_{i,j} = \mathbf{u}_i^T\mathbf{J}_{A,B}\mathbf{v}_j$ such that $\lambda_i \neq \mu_j$, i.e., eigen-values that differ. Thus, we should use a filter that kills off these signals corresponding to eigen-values that do not match. We choose the filter $w(\lambda_i, \mu_j, e) = \frac{1}{(\lambda_i^e - \mu_j^e)^2 + 1}$ instead of $g(\lambda_i)h(\mu_j)$. This filter exactly kills off these signals corresponding to eigen-values that differ (see Figure 4(c) to see an illustration of its action on signals). A note here is that the amplitude and the width of the filter response in Figure 4(c) depends on the spectra of the graphs to be matched as well as on $e$. For simplicity, we denote this filter as $w(\lambda_i, \mu_j)$ in the rest of the paper. Thus, a

---

**Algorithm 1** : 2D-GEM

---

1. **input** An initial correspondence $\mathbf{P}^0$, $\mathbf{U}_k$, $\mathbf{V}_k$, $\mathbf{\Lambda}_k$, $\mathbf{M}_k$, $e$, maximum iteration $iter$, and the LMD error $\{\epsilon_i\}_{i=1}^{iter}$ thresholds.
**while** $0 \leq i \leq iter$ **do**
    2. use one round of the 2-Hop algorithm. i.e., set $t = 1$ in Algorithm 2
    3. use the LMD to locate well anchored pairs (landmarks) $l$
    4. use these well anchored pairs to update the 2D graph spectrum as in (9)
    5. use the updated spectrum for updating the correspondence of poorly matched points $p$:
        $\max_{\mathbf{P}_p} \mathbf{Tr}(\mathbf{P}_p \mathbf{C}_p)$, with $\mathbf{C}_p$ as in (10)
**end while**

---

**Algorithm 2** : 2-Hop

---

1. **input** An initial correspondence $\mathbf{P}$, 2-hop adjacency matrices $\mathbf{A}_2$, $\mathbf{B}_2$, maximum iteration $t$.
**while** $0 \leq i \leq t$ **do**
    2. $\mathbf{P}^{i+1} = GMWM(\mathbf{W})$, where $\mathbf{W} = \mathbf{A}_2 \mathbf{P}^i \mathbf{B}_2$
**end while**
3. **return** $\mathbf{P}^t$

---

noise robust 2D graph convolution based cost matrix for graph matching is given as:

$$\mathbf{C} = \sum_{i,j} w(\lambda_i, \mu_j) \mathbf{u}_i \mathbf{u}_i^T \mathbf{X}_A \mathbf{X}_B^T \mathbf{v}_j \mathbf{v}_j^T. \tag{7}$$

Moreover, to reduce the time complexity, one may use the first $k$ eigen-values $\mathbf{\Lambda}_k = diag(\lambda_i, \cdots, \lambda_k)$, $\mathbf{M}_k = diag(\mu_i, \cdots, \mu_k)$ and eigen-vectors $\mathbf{U}_k = [\mathbf{u}_i, \cdots, \mathbf{u}_k]$, $\mathbf{V}_k = [\mathbf{v}_i, \cdots, \mathbf{v}_k]$, instead of the full spectrum. Doing so, the noise robust 2D graph convolution based cost matrix for graph matching becomes:

$$\mathbf{C} = \sum_{i,j}^{k} w(\lambda_i, \mu_j) \mathbf{u}_i \mathbf{u}_i^T \mathbf{X}_A \mathbf{X}_B^T \mathbf{v}_j \mathbf{v}_j^T. \tag{8}$$

Due to space constraints, we refer the reader to Appendix D for the experimental validation of the 2D graph convolution approach for graph matching.

## 4 Our Algorithm

Having formulated the 2D graph convolution-based approach for graph matching as well as redefined noise in the context of graph matching, we now formulate 2D-GEM (2D Graph convolution-based framework for attempting Exact shape Matching) for non-isometric and nearly-isometric 3D shape matching.

### 4.1 2D-GEM

For our framework, given the xyz coordinates of two shapes $\mathcal{M}_1$ and $\mathcal{M}_2$, we build suitable point-wise descriptors $\mathbf{X}_A$ and $\mathbf{X}_B$, as well as graph laplacians $\mathbf{L}_A$ and $\mathbf{L}_B$ using their triangulations. We then initialize the correspondence using the point-wise descriptors. Then at each iteration, we first follow GRAMPA and update the correspondence using the 2-hop neighborhood graph matching algorithm [84], which makes use of the Greedy Maximum Weight Matching (GMWM) with computational complexity $O(n^2 \log n)$ (see Algorithm 2 and Appendix B). The 2-hop adjacency matrices $\mathbf{A}_2$ and $\mathbf{B}_2$ of the shapes $\mathcal{M}_1$ and $\mathcal{M}_2$, respectively, are such that $[\mathbf{A}_2]_{i,j} = 1$ and $[\mathbf{B}_2]_{i,j} = 1$ if $i$ and $j$ are 2-hop neighbors, and 0 otherwise. Second, we use the notion of conservation of geodesic distances [64] to further constrain 2D-GEM, we capture this via LMD[80]. Finally, we adopt the 2D graph convolution in Equation (8) to filter the correspondence $\mathbf{P}$ rather than the feature similarity matrix $\mathbf{X}_A \mathbf{X}_B^T$ in Equation (8), i.e., we use $\mathbf{P}$ rather than $\mathbf{X}_A \mathbf{X}_B^T$ as our 2D graph signals. This can be seen as de-noising the correspondence at each iteration following the state-of-the-art Kernel-Matching method [42].

In details (see Algorithm 1), we initialize the correspondence via SHOT[70] (though other descriptors and algorithms could also be used as in Appendices D, and F), and then at each iteration ($iter$):

- we update the correspondence using the 2-Hop Algorithm (2) with $t = 1$,
- we find $l$ anchor pairs and $p$ non-anchor pairs via LMD (see Section 4.2) and use them to update the 2D graph spectrum as

$$\mathbf{Z}_l = \mathbf{U}_k^T[l]\mathbf{P}_l\mathbf{V}_k[l], \tag{9}$$

  where $\mathbf{U}_k[l]$, $\mathbf{V}_k[l] \in \mathbb{R}^{l \times k}$ and $\mathbf{P}_l \in \mathbb{R}^{l \times l}$ are the eigen-vectors and correspondence of the $l$-anchor pairs (landmarks) found, and $\mathbf{Z}_l$ is the updated $k \times k$ 2D graph spectrum.

- we use the updated spectrum for updating the correspondence of poorly matched points as $\max_{\mathbf{P}_p} \mathbf{Tr}(\mathbf{P}_p\mathbf{C}_p)$, where

$$\mathbf{C}_p = \sum_{i,j}^{k} w(\lambda_i, \mu_j)\mathbf{u}_i[p][\mathbf{Z}_l]_{i,j}\mathbf{v}_j^T[p], \tag{10}$$

  and $\mathbf{u}_i[p], \mathbf{v}_j[p] \in \mathbb{R}^p$ are the eigen-vectors of the non-landmark points $p$.

**Computation complexity.** The complexity for computing $k$ leading eigen-vectors of a $n \times n$ sparse matrix is $O(kn \log n)$. The complexity for checking LMD is $O(n)$ following DIR [80]. The GMWM has a computational complexity of $O(n^2 \log n)$. Thus, our method is of complexity $O((iter)n^2 \log n) = O(n^2 \log n)$, since $iter$ is usually small, e.g., 10 or 60 in our experiments.

## 4.2 Local Mapping Distortion (LMD)

Here, we describe the LMD employed by 2D-GEM to get landmarks. The LMD was introduced by [80] and [79] for nearly-isometric shapes. Let $T : \mathcal{M}_1 \rightarrow \mathcal{M}_2$ be a map function between two shapes. The LMD of the map $T$ at the point $x_i$ is given as follows:

$$\mathcal{D}_\gamma(T)(x_i) = \frac{\sum_{x_j \in \mathcal{B}_\gamma(x_i)} \mathcal{A}_1(j)DE_T(x_i, x_j)}{\sum_{x_j \in \mathcal{B}_\gamma(x_i)} \mathcal{A}_1(j)}, \tag{11}$$

where $\mathcal{B}_\gamma(x) = \{y \in \mathcal{M}_1 \mid d_{\mathcal{M}_1}(x, y) \leq \gamma\}$ is the $\gamma$-geodesic ball of $x$, $\mathcal{A}_1$ is the area element of the mesh of shape $\mathcal{M}_1$, and $DE_T(x, y) = |d_{\mathcal{M}_1}(x, y) - d_{\mathcal{M}_2}(T(x), T(y))|/\gamma$ represents a pair-wise distance distortion of mapping nearby points $x_i$ and $x_j$ to $T(x_i)$ and $T(x_j)$. A smaller value of $\mathcal{D}_\gamma(T)(x_i)$ means a better map continuity of $T$ at the point $x_i$, in other words, the local distance at the point $x_i$ is well preserved. Based on the above definition of LMD, one can check that if $T$ is an isometric map, then $\mathcal{D}_\gamma(T)(x) = 0, \forall x \in \mathcal{M}_1, \gamma > 0$. Conversely, if $\mathcal{D}_\gamma(T)(x) = 0, \forall x \in \mathcal{M}_1$ for some $\gamma > 0$, then $T$ is isometric. We use the LMD to find well matched pairs, i.e., pairs for which $\{(x_i, T(x_i)) | \mathcal{D}_\gamma(T)(x_i) \leq \epsilon_i\}$, where $\epsilon_i$ is a threshold. We call them anchor points (landmarks $l$), and others non-anchor points (poor-points $p$). We use the landmarks to update the 2D graph spectrum $\mathbf{S}_{kl}$, which we use to update the correspondence of non-landmark points (see Section 4.1). It is important to note that at every iteration we gradually decrease the threshold following Xiang et al. [80], and hence not only is the correspondence $T$ updated, but also the landmarks $l$. For non-isometric shapes we relax this constraint by setting the thresholds $\epsilon_i$ arbitrarily large, such as 100 for 2D-GEM in the experiments (see Section 5.5).

## 4.3 Relationship Between 2D-GEM and Functional Maps

Here we show the relationship between 2D-GEM and the popular Functional Map [56]. The notion of Functional Maps [56] was introduced in order to solve the notoriously difficult QAP resulting from *pair-wise alignment*. A Funcitonal Map is a map between functional spaces $P : L^2(\mathcal{M}_1) \rightarrow L^2(\mathcal{M}_2)$, which can be discretized as an $n \times n$ matrix $\mathbf{P}$ (assuming that each of the shapes $\mathcal{M}_1$ and $\mathcal{M}_2$ has $n$ vertices). Given a pair of orthonormal basis $\mathbf{U} = [\mathbf{u}_1, \cdots, \mathbf{u}_n]$ and $\mathbf{V} = [\mathbf{v}_1, \cdots, \mathbf{v}_n]$ for $L^2(\mathcal{M}_1)$ and $L^2(\mathcal{M}_2)$, respectively, we have $\mathbf{P} = \mathbf{U}\mathbf{S}\mathbf{V}^T$, where $\mathbf{S}$ acts as a basis transformation matrix. On the other hand, recalling 2D-GEM's cost matrix for iteratively de-noising the Permutation matrix in Algorithm 1, i.e.,

$$\mathbf{C} = \sum_{i,j}^{k} w(\lambda_i, \mu_j)\mathbf{u}_i\mathbf{u}_i^T\mathbf{P}\mathbf{v}_j\mathbf{v}_j^T = \sum_{i,j}^{k} w(\lambda_i, \mu_j)\mathbf{u}_i\mathbf{s}_{i,j}\mathbf{v}_j^T, \tag{12}$$

where $s_{i,j} = \mathbf{u}_i^T \mathbf{P} \mathbf{v}_j$. This means that one can interpret the matrix $\mathbf{S}_k \in \mathbb{R}^{k \times k}$ containing the $s_{i,j}$'s as a low-pass approximation of the permutation matrix $\mathbf{P}$ in the truncated Laplacian eigen-basis, i.e., $\mathbf{P} \approx \mathbf{U} \mathbf{S}_k \mathbf{V}^T$ [42]. Equation (12) can thus be interpreted as applying a graph matching-suitable filter to the functional map matrix $\mathbf{S}_k$ [56, 42]. 2D-GEM 1 can thus be regarded as denoising the functional map in the graph laplacian basis while enforcing the preservation of neighborhood using 2-hop witnesses [84], as well as ensuring the preservation of geodesic distances (distortions) via the LMD [80]. This interpretation leads to the beautiful connection between the 2D-graph spectrum and the functional map. Essentially they are the same when we take the 2D-graph signals as the columns of the correspondence matrix. As such the 2D-graph convolution approach is intricately connected to methods such as ZoomOut [54], ICP [4] and BCICP [58], which use the functional map together with the truncated Laplacian eigen-basis to update the correspondence $\mathbf{P}$. Moreover, recent works such as Structured Regularization of Functional Map Computations [59], and DIR[80] tried to address some problems with the traditional functional map approaches (namely, (a) penalizing the functional map update only when it fails to commute with the LBO's of the source and target shapes often fails to produce a good functional map, (b) accumulation of errors in the map update due to poor initial correspondence) by introducing regularizers in the functional map update step. Following these works, 2D-GEM can also be seen as a framework for further constraining the functional map and the point-wise map updates using the LMD and the 2-hop neighborhood witnesses. Another line of work, namely Fast Sinkhorn Filters [57] tried to explain functional map based approaches as spectral alignment regularized algorithms, and as such, 2D-GEM can also be viewed as a spectral alignment regularized algorithm.

## 5  Experiments

We report experimental results that validate the effectiveness and efficiency of 2D-GEM on exact matching of non-isometric and nearly-isometric 3D shapes.

### 5.1  Experimental Set-up

Time comparison between 2D-GEM and GRAMPA is conducted in Matlab on a Windows 10 system with 16GB RAM and Intel(R) i5 11400 CPU @ 2.60-4.4GHz.

### 5.2  Datasets

We conduct experiments to evaluate the performance of our method on two nearly isometric benchmark datasets TOSCA [10], and SCAPE [5], as well as a benchmark non-isometric dataset SHREC'16 (TOPKIDS) [42]. TOSCA consists of 80 shapes in 8 different categories (human and animal shapes) with vertex numbers ranging from 4k to 50k. SCAPE has 71 shapes (12,500 vertices for each) of the same person with different poses. TOPKIDS contains 25 shapes of the same class with up to 12K vertices, undergoing near-isometric deformations in addition to large topological shortcuts (noise).

### 5.3  Evaluation Metrics

We use the geodesic error as our error metric [20]. Given that the correspondence of an algorithm maps $x \in \mathcal{M}_1$ to $y \in \mathcal{M}_2$, and the true correspondence maps $x$ to $y^*$, the geodesic error is defined as $e(x) = \frac{d_{\mathcal{M}_2}(y,y^*)}{diam(\mathcal{M}_2)}$, where $d_{\mathcal{M}_2}$ denotes the geodesic distance on $\mathcal{M}_2$, and $diam(\mathcal{M}_2)$ is the geodesic diameter of $\mathcal{M}_2$.

### 5.4  Baselines

For non-isometric shape matching, i.e., for matching on the TOPKIDS dataset, we compare 2D-GEM with the following methods: EM [62], GE [41], RF [61], PFM [60], FSPM [49], Kernel-Matching [42], and GRAMPA [20].

For near-isometric shape matching, i.e., for matching on the SCAPE and TOSCA datasets, we compare 2D-GEM to the following methods: SGMDS[2], FM[56], BIM[34], Mobius[48], Best-Conformal [34], Kernel-Matching [42], DIR-500 [80] which uses 500 eigen-vectors, DIR-1000 [80]

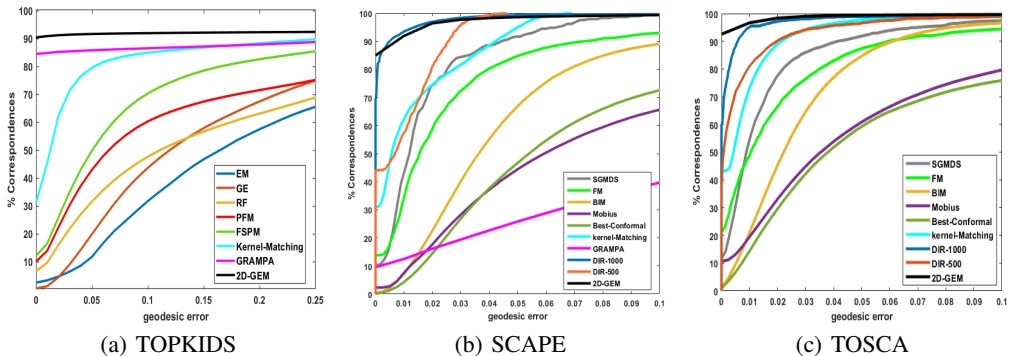

(a) TOPKIDS      (b) SCAPE      (c) TOSCA

Figure 5: Performance comparison on TOPKIDS 5(a), SCAPE 5(b), and TOSCA 5(c).

which uses 1000 eigen-vectors, and GRAMPA (on SCAPE, not on TOSCA due to its time complexity and the fact that 2D-GEM significantly outperforms it as shown on TOPKIDS and SCAPE).

## 5.5 Parameter Settings

**For 2D-GEM**, we initialize the correspondence by the GMWM on SHOT [70], and use $e = 10$ (see Appendix D for more analysis on $e$). For nearly-isometric shapes (TOSCA and SCAPE), we use 500 eigen-vectors of the LBO. Following Xiang et al. [80], we use the second ring neighborhood size for LMD criterion and set the maximum iteration number to $iter = 10$. We also set the LMD thresholds, we use $[0.5280, 0.4560, 0.3840, 0.3120, 0.2400, 0.1680, 0.1680, 0.1680, 0.1680, 0.1680]$. For non-isometric shapes (TOPKIDS), we set the thresholds to 100 for all iterations, use $iter = 60$, and 10 LBO eigen-vectors.

**For other baselines**, we report their results as reported in their papers i.e., we use their curves. Except for GRAMPA where we re-run using the 2-hop improvement algorithm instead which performs much better than the 1-hop, i.e., the adjacency. For fair comparison with 2D-GEM , we set $t = 60$ for this improvement on GRAMPA.

## 5.6 Performance Analysis

**Comparison on non-isometric shapes.** [2] GRAMPA is the state-of-the-art non-isometric shape matching method that achieves an excellent performance on the exact matching task on the noisy dataset TOPKIDS (Figure 5(a)). It achieves $84.6\%$ on this dataset at geodesic error 0. This shows that GRAMPA is very robust to noise as expected (note that this performance is much higher than the $60\%$ reported in their original paper because we used the 2-hop adjacency instead of the 1-hop for refinement as explained in Appendix B.2). However, our method 2D-GEM further outperforms GRAMPA and all other methods, including GRAMPA at geodesic error 0. As reported in Figure 5(a), where 2D-GEM achieves a $90.5\%$ accuracy. Moreover, Figures 2(a) and 2(b) show that 2D-GEM is significantly faster than GRAMPA since 2D-GEM only needs to compute a subset of the eigen-vectors, unlike GRAMPA which needs the whole spectrum and thus has a complexity of $O(n^3)$.

**Comparison on nearly-isometric shapes.** DIR is the state-of-the-art model for nearly-isometric shape matching. DIR outperforms all the other baselines except 2D-GEM. The better DIR-1000 using 1000 eigen-vectors achieves $69.5\%$ accuracy at geodesic error 0 on SCAPE, and $59.8\%$ accuracy at geodesic error 0 on TOSCA. Using the LMD proposed for the DIR model by Xiang et al. [80] for extra constraints on 2D-GEM, 2D-GEM not only generalizes to nearly-isometric shapes, but it achieves the state-of-the-art results using only 500 eigen-vectors, outperforming even DIR-1000. 2D-GEM achieves $92.5\%$ accuracy (outperforming DIR-1000 by more than $40\%$) at geodesic error 0 on the TOSCA dataset, as well as $84.9\%$ accuracy on the SCAPE dataset (outperforming DIR-1000 by almost $20\%$) . Note that these 2D-GEM's accuracy improves using more eigen-vectors or increasing $iter$, but this also increases the running time. Moreover, 2D-GEM's gain over GRAMPA (the state-of-the-art non-isometric shape matching method) can also be seen to extend to nearly-isometric shapes. As noted earlier in Section 1, GRAMPA does not generalize to nearly-isometric shapes since the

---

[2]Ablation studies on 2D-GEM in Appendix E. Code at: `https://github.com/BarakeelFanseu/2D-GEM`

assumptions for nearly-isometric shape matching are different from those of non-sometric shape matching given that isometries preserve geometric properties which are lost by non-isometries. This is seen by its poor performance on the SCAPE dataset (Figure 5(b), where even for exact matching, i.e., at geodesic error 0, it only achieves around $10\%$). In contrast, 2D-GEM (using the LMD for extra constraints) generalizes to nearly-isometric shapes and achieves the state-of-the-art performance. We do not attempt to use the LMD on GRAMPA since (1) it is not clear how to use GRAMPA in an iterative way as it is usually only used as an initialization for other algorithms (see Appendix B), (2) 2D-GEM outperforms GRAMPA even without the LMD as shown on the noisy TOPKIDS dataset (Figure 5(a)), and (3) 2D-GEM with or without the LMD is still faster than GRAMPA without the LMD (see Figures 2(a) and 2(b)).

## 5.7 Limitations of 2D-GEM

There are three main limitations to this work. First, for nearly-isometric shapes (shown in Figure 3(a)), 2D-GEM sometimes produces maps that are not continuous (smooth), though much more accurate than the other baselines. Second, although the time complexity $O(n^2 \log n)$ of 2D-GEM is much lower than the $O(n^3)$ of GRAMPA, it is much worse than the $O(n \log n)$ of other methods such as ZoomOut and DIR. Lastly, unlike GRAMPA, we do not provide rigorous theoretical exact recovery guarantees for 2D-GEM on random graph matching.

## 6 Conclusions

We presented how the 2D graph convolution could be used for graph matching. Using this 2D graph convolution, we re-defined noise in the context of graph matching and proposed a model (2D-GEM) that addresses the task of exact 3D shape matching on non-isometric (noisy) datasets as well as on nearly-isometric datasets. We then showed that 2D-GEM significantly outperforms existing methods on both non-isometric shapes and nearly-isometric shapes. For future work, we propose to address two key limitations of 2D-GEM, i.e., further reducing the time complexity of $O(n^2 \log n)$ and providing theoretical guarantees on its exact recovery performance. Moreover, we also propose to explore the generalization of 2D-GEM for correspondence tasks that allow using graph representation such as partial shape matching, image matching, and network de-anonymization. Besides, it is also promising to consider the potential distribution shifts [12], as well as the optimization trade-offs [13].

## 7 Broader Impact

Given that our method needs no training, is unsupervised and based on intrinsic shape properties, it can positively benefit the industry and society as a whole in that; (a) it may be easily generalized, and most importantly (b) it needs no human labels nor training, which can help reduce the cost and time taken for the shape correspondence problems in different crucial fields such as medicine, and computer graphics.

## 8 Acknowledgements

We thank the reviewers for their valuable comments. This work was partially supported by GRF 14208318 from the RGC of HKSAR and the CUHK direct grant 4055146. Lin Zhang and Bo Li were supported in part by RGC RIF grant R6021-20 and RGC GRF grant under contracts 16209120 and 16200221. Bo Han was supported by NSFC YSF No. 62006202, RGC ECS No. 22200720, and Guangdong Basic and Applied Basic Research Foundation No. 2022A1515011652.

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
