# OpenReview forum: "Exact Shape Correspondence via 2D graph convolution"
_NeurIPS.cc/2022/Conference — NeurIPS 2022 Accept_

### Official Review · Reviewer_HJ7U · 2022-07-01

**Rating:** 6
**Confidence:** 4
**Soundness:** 3 good
**Presentation:** 3 good
**Contribution:** 3 good

**Summary:**

This paper proposes a novel method for exact shape correspondence based on 2D graph convolution. The main idea is to adapt graph convolution and signal denoising on graphs to the task of shape matching. The proposed pipeline exploits a specific filter inspired by isomorphic graph matching to achieve this goal. The matching results both in the case of near-isometric and non-isometric shapes are accurate. The timing is comparable to and generally faster than the most similar competitors. Furthermore, it reaches a more significant percentage of points for which the geodesic error is 0 compared to alternative approaches.

**Questions:**

1 - Why, in line 164, is the text referring to z_{i,j} while the discussion is on a single graph and its filter g? It should be z_i, or am I missing anything?

2 - Is Figure 4 (c) generated on two isomorphic graphs? From the shape of the filter, it seems to be the case. If it is the case, it should be reported at least in the caption because this filter is dependent on the graphs (the spectra).

3 - The functional map framework [51] proposes a constraint for shape matching (commutativity with the Laplacian) that corresponds to a mask at least similar to the one suggested as a spectral filter in this paper. The recent work "Structured Regularization of Functional Map Computations" by J. Ren et al. (SGP 2019) provides more details on this constraint. Can be any further connection analyzed?

4 - What is the quality of the initialization provided by the proposed pipeline? Adding the evaluation of the initialization (both quantitatively to the curves and qualitatively to the images) will provide a general idea of its quality and highlight the contribution of the proposed pipeline.

5 - Selecting the threshold equal to 100 in the LMD for non-isometric pairs is similar to deactivating this step in the procedure. Is it correct? Or, with this threshold, is the LMD procedure removing any (probably few) matches with a huge error? This last claim seems confirmed by the ablation study in Figure 7 case (3), but this discussion should appear in the main paper.

**Limitations:**

I do not see any potential negative societal impact. The limitations listed by the authors give a complete analysis of the open problems in the proposed approach. One possible missing scenario is the one in which the two shapes have a completely different discretization. SHREC19 "Matching Humans with Different Connectivity" is a possible benchmark to evaluate the performance of the proposed method in this specific case.
Moreover, some more challenging settings can be discussed, such as partialities, point clouds, and noisy data from real acquisitions. In these cases, where geodesic constraints

**Strengths And Weaknesses:**

Strengths:
- The connection between 2D graph convolution and shape matching is interesting.
- The paper provides all the necessary background to understand the proposed idea.
- The performance on the datasets involved in the evaluation is good, and the proposed pipeline improves the state-of-the-art of the competitors considered without training and ground truth data.

Weaknesses:
- Section 4 is not easy to follow. An image of the pipeline with a visualization of the different steps can help improve the text's clarity.
- I am not sure about the novelty of the 2D graph convolution as a filter for shape matching. Section 3 should start with a more explicit and precise statement about the originality. Furthermore, a more comprehensive discussion about the connection with related works is missing in the current version. The paper can benefit from a specific paragraph on this discussion that highlights the novelty of the proposed method but in comparison to existing techniques.
- The pipeline proposed in Section 4 seems a bit incremental. It is given by SHOT+GMWM+[LMD]+refinement. The last part (the last step of the procedure from Algorithm 1) appears to be the main contribution of this paper. If I am not missing something, the first part of the complete pipeline can be composed of existing methods with some modifications on the choice of the parameters to target different settings. This discussion does not invalidate the submission, but the current version of the manuscript should be modified accordingly if correct. If I am wrong, the current version requires a more detailed motivation on the selected solution for step 1,2,3,4 in the Algorithm to clarify why all the procedure is a valid contribution from this work.

Some minor points:
- In line 113, adding a reference to the normalized symmetric Laplacian at the beginning of section 2 can make the paper self-contained.
- In line 167, \mu_i should be \mu_j.
- In Figure 4, fewer values on the x and y axes can make the figure clearer.
- Around line 178, a description of e should be given. From the caption of Figure 4 and section 5, it is possible to deduce that it is a real number (positive? integer?), but it could help to have it explicitly defined in the text.
- Reference [53] contains a typo: Rodolì should be Rodolà.
- The text in Figure 2 is too small. The quality of this figure should be improved by using a tikz file or other alternatives. Same for Figures 3 and 5.

---

> ### Author Response · Authors · 2022-08-02
> **Responses to Reviewer HJ7U**
>
> **Question 1.** Why, in line 164, is the text referring to $z_{i,j}$ while the discussion is on a single graph and its filter g? It should be $z_i$, or am I missing anything?
>
> **Response.** Thank you for the careful reviews. We have corrected this typo as well as other typos mentioned in the review.
>
> **Question 2.** Is Figure 4 (c) generated on two isomorphic graphs? From the shape of the filter, it seems to be the case. If it is the case, it should be reported at least in the caption because this filter is dependent on the graphs (the spectra).
>
> **Response.** The two graphs are not isomorphic. However, whether the graphs are isomorphic or not, the shape of the filter remains the same. Though as pointed out by the reviewer, the height and the width of the response in the plots will be affected by the parameter $e$ and the spectra of the two graphs. As suggested by the reviewer, we added a note of the plot in the paper.
>
> Question 3. The functional map framework [51] proposes a constraint for shape matching (commutativity with the Laplacian) that corresponds to a mask at least similar to the one suggested as a spectral filter in this paper. The recent work "Structured Regularization of Functional Map Computations" by J. Ren et al. (SGP 2019) provides more details on this constraint. Can be any further connection analyzed?
>
> **Response.** [1]proposed additional regularizers to update the functional map (namely, a descriptor-commutative regularizing term, an orientation-preserving term, and a resolvent term). [1] proposed these regularizers because only adding a penalty when the map is not commuting with the Laplace-Beltrami operators on both the source and target shapes may fail in some cases (as pointed out in [1]). 2D-GEM may be used to  bridge the gap between the popular Functional Map and the 2D-Graph convolution. In this regard, 2D-GEM can also be interpreted as a regularised spectral alignment method to update the map. Thus, it is related to [1] with the difference that the regularizers in our case are neighborhood preservation (enforced via the 2-Hop) and distortion preservation (enforced via the LMD) in addition to the fact that the map in our case is computed in the spectrum of the graphs associated with the shape's triangulation rather than in the traditional shape LBO's spectrum. This discussion was added to the paper in Appendix A.2 (lines 677-700).
>
> **Question 4.** What is the quality of the initialization provided by the proposed pipeline? Adding the evaluation of the initialization (both quantitatively to the curves and qualitatively to the images) will provide a general idea of its quality and highlight the contribution of the proposed pipeline.
>
> **Response.**  As shown in Appendix D, case (2) 2D-convolution is a good initializer itself. In fact, 2D-GEM is comparable with GRAMPA when there are no features. Moreover, as shown by cases (3)-(6) when informative features are present, the 2D-convolution provides an even better initialization. Finally, in the added Appendix F that reports the results of different initializations, we have DSHoT which is the initialization by 2D-GEM itself using SHOT descriptors, i.e., (a) directly using SHOT to build the similarity matrix (i.e., computing XaXb'), (b) using 2D-GEM as in Equation 4 to initialize the map, and (c) using this initialized map in the algorithm in Section 4.
>
> **Question 5.** Selecting the threshold equal to 100 in the LMD for non-isometric pairs is similar to deactivating this step in the procedure. Is it correct? Or, with this threshold, is the LMD procedure removing any (probably few) matches with a huge error? This last claim seems confirmed by the ablation study in Figure 7 case (3), but this discussion should appear in the main paper.
>
> **Response.** Yes, setting the LMD so high on non-isometric shapes is to deactivate the step. This is similar to asking the number of anchor points selected to be large since most points' LMD will be below the threshold. This can be seen as a relaxation of the constraint of enforcing the conservation of geodesic properties. We added Appendix H to the paper to show more sensitivity studies on the LMD thresholds.
>
>
> ***
> **References**
>
> [1] Ren, J., Panine, M., Wonka, P., and Ovsjanikov, M. Structured regularization of functional map computations. Comput. Graph. Forum 38, 5 (2019), 39–53.

---

> > ### Author Response · Authors · 2022-08-02
> > **Responses to Reviewer HJ7U**
> >
> > **Limitations 1.** I do not see any potential negative societal impact. The limitations listed by the authors give a complete analysis of the open problems in the proposed approach. One possible missing scenario is the one in which the two shapes have a completely different discretization. SHREC19 "Matching Humans with Different Connectivity" is a possible benchmark to evaluate the performance of the proposed method in this specific case. Moreover, some more challenging settings can be discussed, such as partialities, point clouds, and noisy data from real acquisitions. In these cases, where geodesic constraints
> >
> > **Response.** These are indeed open and challenging scenarios. Due to the time constraint at this point we only briefly mentioned these challenging scenarios and datasets in Appendix F. However, we will endeavour to have a broader study of these challenging scenarios and try to add a discussion section to the appendix. Thank you so much again for your insightful comments.

---

> > ### Comment · Reviewer_HJ7U · 2022-08-03
> > **Some fixing and details on the main paper are required.**
> >
> > Dear authors thank you for the accurate answer to my review.
> >
> > I appreciate the discussion on related papers: [1] in answer to my review and [1] in reply to NtNA's review. I suggest briefly adding them to the related work discussion in Section 1. In the new version submitted, I cannot see them.
> >
> > The value of the LMD threshold is quite essential to me and should be pointed out in section 4.2, referring to the wider discussion in the supplementary materials.
> >
> > Reading the other reviews, I found two main weaknesses recognized by more than one reviewer and not well addressed by the submitted answers.
> > 1) I am unsure about the novelty of the 2D graph convolution as a filter for shape matching. Section 3 should start with a more explicit and precise statement about the originality. Furthermore, a more comprehensive discussion about the connection with related works is missing in the current version. The paper can benefit from a specific paragraph on this discussion that highlights the novelty of the proposed method but in comparison to existing techniques. Similar comments were given in other forms by all the other reviewers, but I am missing a deep discussion on that.
> > 2) An analysis of the initialization's quality is a piece of crucial information and should not be limited to the supplementary. I think it can be added both qualitatively and quantitatively in the paper's figures (e.g., figure 1). A similar and strongly related comment was shared by NtNA reviewer at least.
> >
> > I would suggest some further minor fixing to the paper:
> > - reference [57] (that was [53]) contains the name Rodolì instead of Rodolà and has not been fixed.
> > - text in Figure 2 and in other figures is too small.

---

> > > ### Author Response · Authors · 2022-08-09
> > > **Thanks for further Comments to Reviewer HJ7U**
> > >
> > > Thank you again for your comments and suggestions.
> > >
> > > >Q.1. I appreciate the discussion on related papers: [1] in answer to my review and [1] in reply to NtNA's review. I suggest briefly adding them to the related work discussion in Section 1. In the new version submitted, I cannot see them.
> > >
> > > R.1. We will discuss in Section 1 that Fast Sinkhorn Filters [1] and Structured Regularization of Functional Map Computations [2] are closely related to 2-D GEM. First, the Meta algorithm from [1] enables us to view 2D-GEM as a spectral alignment regularized algorithm where two regularization constraints of (i) neighborhood preservation and (ii) Local Map Distortion are enforced depending on the nature of the shape (i.e., whether isometric or non isometric).  Second, the Fast Sinkhorn Filter subsequently proposed could be used  to replace the GMWM in the 2D-GEM framework. However, the two constraints used in 2D-GEM (neighborhood and map distortion preservation) are different from the regularizer used in [1] to ensure that the optimization stays convex. Moreover, the motivation of 2D-GEM stems from spectral graph theory while that of [1] stemmed from the popular Regularized Optimal Transport.  Third, Structured Regularization of Functional Map Computations [2], like 2D-GEM, is a regularized spectral alignment method. It was motivated by the fact that having an unregularized update for the map in the spectral domain (following the popular Functional Map [3]) can often fail. To address this, [2] proposed three regularization terms for the update, namely (i) a descriptor-commutative regularizing term, (ii) an orientation-preserving term, and (iii) a resolvent term. Due to the page limit, currently we have only added [1] and [2] as citations in Section 1 together with other closely related works such as the popular Functional map framework previously pointed out by the Reviewer. We will include the above detailed discussion of [1] and [2] in the paper when an extra page is granted if the paper is accepted.
> > >
> > > >Q.2. I am unsure about the novelty of the 2D graph convolution as a filter for shape matching. Section 3 should start with a more explicit and precise statement about the originality. Furthermore, a more comprehensive discussion about the connection with related works is missing in the current version. The paper can benefit from a specific paragraph on this discussion that highlights the novelty of the proposed method but in comparison to existing techniques. Similar comments were given in other forms by all the other Reviewers, but I am missing a deep discussion on that.
> > >
> > > R.2.   We have added a concise statement about the originality of 2-D graph convolution as the first sentence of Section 2 where we discuss the connection of our work with traditional graph convolution: To the best of our knowledge, we are the first to propose a 2-D graph convolution for graph alignment and shape matching. The most closely related work is the work that uses traditional convolution for graph alignment [20]. Traditionally, GCN models learn point embeddings for each node (i.e., points on the graphs individually, with the help of the convolution based neural networks that have been shown to work well on graphs as they act as lowpass filters). To learn these embeddings, GCN models employ a suitable pointwise alignment cost to get the map. In contrast, we show that the alignment problem can be naturally reduced to a 2D graph convolution, and then we propose to redefine noise in the context of graph matching, which leads to the proposed filter. We have validated this approach on random graphs (Appendix C and D) as well as on shapes (Section 5). We will add the above detailed discussion to the final version of the paper.
> > >
> > > >Q.3. An analysis of the initialization's quality is a piece of crucial information and should not be limited to the supplementary. I think it can be added both qualitatively and quantitatively in the paper's figures (e.g., figure 1). A similar and strongly related comment was shared by NtNA Reviewer at least.
> > >
> > > R.3. We have done extensive studies on the initializations in Appendix H as well as the LMD’s influence on 2D-GEM in Appendix G. We found that 2D-GEM is generally robust to initializations. Though we need to rerun the experiments on the non-isometric shapes for the HKS and WKS, since we only used 100LBOs which may be the cause for the poor performance, besides the well-known fact that they are not very robust to noise.  We added a brief comment on this in Section 4. We will add the above discussion and move some of the detailed analysis from Appendices H and G to the main content in the final version of the paper. Thank you again for your suggestion.
> > >
> > > >Q.4. I would suggest some further minor fixing to the paper:
> > > reference [57] (that was [53]) contains the name Rodolì instead of Rodolà and has not been fixed.
> > >
> > > R.4. We have updated the reference. Figures were also enlarged. Thank you again for your careful reading.

---

### Official Review · Reviewer_NtNA · 2022-07-07

**Rating:** 5
**Confidence:** 4
**Soundness:** 3 good
**Presentation:** 3 good
**Contribution:** 2 fair

**Summary:**

This paper proposes a new shape matching method called 2D-GEM, based on 2D graph convolution. The method is straight forward: at each iteration, (1) first find a set of reliable correspondences with small enough local smoothness error (called LMD); (2) use these reliable correspondences to compute/update a k-by-k graph spectrum; (3) use the updated graph spectrum to compute the correspondences for poorly matched points.

The proposed method achieves the state-of-the-art performance on shape matching on SHREC'16 TOPKIDS dataset and original FAUST/TOSCA dataset; and on graph matching on Erdos Regny random graphs.

**Questions:**

- As mentioned above, what's the performance of the proposed method on remeshed FAUST/SCAPE with SHOT initialization and other types of initializations (e.g., wave kernel signatures).
- What's the key difference between 2D-GEM and "Fast Sinkhorn Filters": it seems both methods formulate the matching problem as a linear assignment in the spectral space. (of course, 2D-GEM uses a specially chosen filter and only update the correspondences between non-anchor points) it would be nice to know the critical differences in formulation and it might be better to include such a discussion, since: (1) the  2D graph spectrum computed in Eq.(9) seems to be equivalent to computing a functional map using the anchor points correspondences; (2) the correspondences updating in Eq. (10) seems similar to the linear assignment based on the computed functional map formulated in the Fast Sinkhorn Filters.
- How are the thresholds in L272 chosen? The scales of the thresholds used for different datasets are very different. Is the algorithm sensitive to thresholds?
- What are the heuristics to choose the number of LBO? Why it is enough to use 10 LBO for TOPKIDS, but 500 LBO for FAUST/SCAPE? Does the algorithm work for 10 LBO for FAUST/SCAPE dataset? Also why 500 LBO is used for the ablation study on TOPKIDS while 10 LBO is used for generating results?

**Limitations:**

The limitations are adequately addressed. However, it might be worth mentioning the parameter tuning as well since it seems non-trivial.

**Strengths And Weaknesses:**

*Clarity*:
- (++) the paper is well written and easy to follow. The provided code also helps to better understand the algorithm.

*Originality*
- (++) The proposed method 2D-GEM is a generalization of the GRAMPA method: the mesh Laplacian is used instead of the graph adjacency; it can also incorporate per-vertex features; a different filter is used.
- (--) comparison to GRAMPA: though the differences between 2D-GEM and GRAMPA are briefly discussed in Appendix B.3., the novelty seems to be limited. For example, though 2D-GEM is generalized to handle per-vertex features, in the real algorithm, no features are used. Specifically, as described in L205, the correspondence P is used as graph signals (instead of the pre-built descriptors XaXb). It sounds like the newly introduced feature is useless. Also according to the ablation study, the most important or performance-dominant part of the algorithm is the 2-hop initialization, which is proposed by Yu et. al. [77].

*Quality*
- (++) 2D-GEM achieves higher accuracy on TOPKIDS/FAUST/TOSCA with initializations computed from SHOT descriptors.
- (++) 2D-GEM is faster than GRAMPA since in each iteration not all vertices are updated, which leads to less and less number of non-anchor points for linear assignment
- (--) It has been known that the SHOT descriptors can overfit to the uniform triangulation as used in the original FAUST/SCAPE dataset (For example, see discussion in "DiffusionNet: Discretization Agnostic Learning on Surfaces"). A more convincing justification would be testing the algorithm (and all the baselines) on remeshed or resampled dataset with different types of initializations (as done in ZoomOut [49]).

---

> ### Author Response · Authors · 2022-08-02
> **Responses to Reviewer NtNA**
>
> **Question 1.** As mentioned above, what's the performance of the proposed method on remeshed FAUST/SCAPE with SHOT initialization and other types of initializations (e.g., wave kernel signatures)?
>
> **Response.** We added this to the paper in Appendix F. We note that we used a sample shape from TOSCA re-meshed, 500LBOs for 2D-GEM and 100 for the HKS and WKS respectively and set the LMD to 100 for all 60 iterations. Though the results are not as good as expected, 2D-GEM is still stable to initializations and still competitive with the other Baselines. Due to the time limit, we have not been able to properly amend the framework to remeshed shapes given their different dynamics, which significantly affect the triangulation that is a key component for 2D-GEM (as they are used for the 2D-graph convolution as well as for the 2-Hop algorithm).  Due to the the fact that open-review does not allow us to upload figures, we kindly refer the reviewer to the newly added Figures~9 and 10 in the paper.
>
> **Question 2.** What's the key difference between 2D-GEM and "Fast Sinkhorn Filters": it seems both methods formulate the matching problem as a linear assignment in the spectral space. (of course, 2D-GEM uses a specially chosen filter and only update the correspondences between non-anchor points) it would be nice to know the critical differences in formulation and it might be better to include such a discussion, since: (1) the 2D graph spectrum computed in Eq.(9) seems to be equivalent to computing a functional map using the anchor points correspondences; (2) the correspondences updating in Eq. (10) seems similar to the linear assignment based on the computed functional map formulated in the Fast Sinkhorn Filters.
>
> **Response.** Thank you for pointing our the work "Fast Sinkhorn Filters" [1] to us. [1]  provides a unifying framework for spectral methods. Following the Meta Algorithm from [1], 2D-GEM can equally be seen as a spectral alignment regularised algorithm. [1] proposed the Sinkhorn filter instead of the KNN and Auction (and GMWM) matching. It is worth considering to incorporate the sinkhorn filter into 2D-GEM. Concerning the differences, besides the difference in the motivation for Fast Sinhorn Filters (Regularized optimal transport) and 2D-GEM (spectral graph theory), the main differences between them are first the fact that the cost matrix proposed in Fast Sinkhorn Filters is a distance matrix (euclidean) between the eigen-vectors, while that in 2D-GEM is based on the spectrum of the graphs of the corresponding shapes. Second, rather than using single eigen-vectors on each shape, 2D-GEM uses pairs of eigen-vectors for its cost matrix. Finally, the regularizers used for 2D-GEM enforce both neighbourhoods preservation (via the 2-Hop algorithm) and geodesic distortion preservation (via the LMD), while the regularizer in fast Sinkhorn Filters is mainly to make the problem strictly convex which gives the interpretation of a regularized transport plan. This discussion was added to the paper in Appendix A.2 (lines 677-700).
>
> **Question 3.** How are the thresholds in L272 chosen? The scales of the thresholds used for different datasets are very different. Is the algorithm sensitive to thresholds?
>
> **Response.** We refer the reviewer to the added sensitivity study on the LMD thresholds for 2D-GEM in Appendix H in the paper. As discussed in Section~1 (Introduction) and Section 4.2 (LMD), LMD is used as an extra constraint to enforce the conservation of geodesic distortions when updating a map. Therefore, one can adjust the thresholds depending on whether the dataset and the shapes are isometric or not (i.e., setting low thresholds enforces the LMD constraints more, while setting higher thresholds relaxes it). That is why on TOPKIDS which is a non-isometric dataset with high noise levels we set the thresholds to 100, thereby effectively relaxing the LMD constraint; while on TOSCA and SCAPE we set the thresholds much lower following the previous works SEQA[2] and DIR, thus enforcing the LMD constraint.
>
>
>
>
> ***
> **References**
>
> [1] Pai, G., Ren, J., Melzi, S., Wonka, P., and Ovsjanikov, M. Fast sinkhorn filters: Using matrix scaling for non-rigid shape correspondence
> with functional maps. In Proceedings of the IEEE/CVF Conference on Computer Vision and Pattern Recognition (CVPR) (June 2021), pp. 384–393.
>
> [2]  Xiang, R., Lai, R., and Zhao, H. Efficient and robust shape correspondence via sparsity-enforced quadratic assignment. In Proceedings ofthe IEEE/CVF Conference on Computer Vision and Pattern Recognition (2020), pp. 9513–9522.

---

> > ### Author Response · Authors · 2022-08-02
> > **Responses to Reviewer NtNA**
> >
> > **Question 4.** What are the heuristics to choose the number of LBO? Why it is enough to use 10 LBO for TOPKIDS, but 500 LBO for FAUST/SCAPE? Does the algorithm work for 10 LBO for FAUST/SCAPE dataset? Also why 500 LBO is used for the ablation study on TOPKIDS while 10 LBO is used for generating results?.
> >
> > **Response.** Generally, as shown in Appendix D on random graphs, an increase in the number of eigen-vectors used should lead to an increase in performance of the 2D-graph convolution. However, this also increases the computational cost, and thus users are encouraged to find the best trade-off depending on their needs.  We refer the reviewer to the initialization plots in Figures 9 and 10 in the paper where we used 500LBOs on all shapes (isometric, non-isometric and re-meshed), which gives similar or better performance than those reported in the Experiments (Section 5). though indeed for these plots we used a single shape from each dataset rather than the we datasets. We equally added Appendix H in the paper to show the sensitivity of 2D-GEM to different choices of LMD thresholds.
> >
> > **Limitations 1.** The limitations are adequately addressed. However, it might be worth mentioning the parameter tuning as well since it seems non-trivial.
> >
> > **Response.** We added Appendix H to show how the LMD were chosen. For the LBOs, generally increasing the number of LBOs used will give better performance, though we will need to trade off computational cost for accuracy.

---

> > > ### Comment · Reviewer_NtNA · 2022-08-03
> > > **some minor suggestions**
> > >
> > > Dear authors,
> > >
> > > Thanks for your detailed answers.
> > >
> > > I agree with Reviewer HJ7U that it would be nice to add a more detailed discussion to these strongly related work in Sec. 1 and in Sec. 3/4 by highlighting the major differences and the novelty (as discussed in the answers).
> > >
> > > Here are some minor suggestions:
> > > 1. L969 - missing reference to equation
> > > 2. Fig. 9-10: thanks for the new figures, it would be more informative to report some stats such as average in caption/legend or use log-scale for y-axis, or show a zoomed-in version, currently the curves in the figures are clustered together.
> > > 3. Fig. 9-10 is reported on a single shape pair: it would be nice and more convincing to use a slightly larger dataset (not necessarily the complete dataset, but at least contains 10-20 shapes), it might also be helpful to show the differences between different features/methods.

---

> > > > ### Author Response · Authors · 2022-08-09
> > > > **Thanks for further comments  to Reviewer NtNA**
> > > >
> > > > Dear reviewer NtNA,
> > > >
> > > > Thanks for your further comments and outstanding suggestions.
> > > >
> > > > >Q1. add a more detailed discussion to these strongly related work in Sec. 1 and in Sec. 3/4 by highlighting the major differences and the novelty (as discussed in the answers)
> > > >
> > > > R.1. We will discuss in Section 1 that Fast Sinkhorn Filters [1] and Structured Regularization of Functional Map Computations [2] are closely related to 2-D GEM. First, the Meta algorithm from [1] enables us to view 2D-GEM as a spectral alignment regularized algorithm where two regularization constraints of (i) neighborhood preservation and (ii) Local Map Distortion are enforced depending on the nature of the shape (i.e., whether isometric or non isometric).  Second, the Fast Sinkhorn Filter subsequently proposed could be used  to replace the GMWM in the 2D-GEM framework. However, the two constraints used in 2D-GEM (neighborhood and map distortion preservation) are different from the regularizer used in [1] to ensure that the optimization stays convex. Moreover, the motivation of 2D-GEM stems from spectral graph theory while that of [1] stemmed from the popular Regularized Optimal Transport.  Third, Structured Regularization of Functional Map Computations [2], like 2D-GEM, is a regularized spectral alignment method. It was motivated by the fact that having an unregularized update for the map in the spectral domain (following the popular Functional Map [3]) can often fail. To address this, [2] proposed three regularization terms for the update, namely (i) a descriptor-commutative regularizing term, (ii) an orientation-preserving term, and (iii) a resolvent term. Due to the page limit, currently we have only added [1] and [2] as citations in Section 1 together with other closely related works such as the popular Functional map framework previously pointed out by the Reviewer. We will include the above detailed discussion of [1] and [2] in the final version of the paper.
> > > > We have also added a concise statement about the originality of 2-D graph convolution as the first sentence of Section 2 where we discuss the connection of our work with traditional graph convolution: To the best of our knowledge, we are the first to propose a 2-D graph convolution for graph alignment and shape matching. The most closely related work is the work that uses traditional convolution for graph alignment [20]. Traditionally, GCN models learn point embeddings for each node (i.e., points on the graphs individually, with the help of the convolution based neural networks that have been shown to work well on graphs as they act as lowpass filters). To learn these embeddings, GCN models employ a suitable pointwise alignment cost to get the map. In contrast, we show that the alignment problem can be naturally reduced to a 2D graph convolution, and then we propose to redefine noise in the context of graph matching, which leads to the proposed filter. We have validated this approach on random graphs (Appendix C and D) as well as on shapes (Section 5). We will add the above detailed discussion to the final version of the paper.
> > > >
> > > > > Q.2. missing reference to equation
> > > >
> > > > R.2. We have corrected this reference mistake, which is in line 982 now.
> > > >
> > > > >Q.3. Fig. 9-10: thanks for the new figures, it would be more informative to report some stats such as average in caption/legend or use log-scale for y-axis, or show a zoomed-in version, currently the curves in the figures are clustered together.
> > > >
> > > > R.3. We have updated the plots and also added notes to them.
> > > >
> > > > >Q.4. Fig. 9-10 is reported on a single shape pair: it would be nice and more convincing to use a slightly larger dataset (not necessarily the complete dataset, but at least contains 10-20 shapes), it might also be helpful to show the differences between different features/methods.
> > > >
> > > > R.4. We are currently re-running the experiments on a larger subset of the datasets as suggested. We have added the results for the larger subset on Isometric and Non Isometric shapes (using a subset of 10 shapes for each). Should those on remeshed shapes come out before the August 9th deadline, we will update the figures too, else we will try to update in the final version of the paper. Moreover, we noticed that for the remeshed shape we had used the LMD 100 for the Tosca Isometric dataset provided by [1], rather than lower LMDs as the dataset is assumed to be isometric (we will correct this in the future updates).
> > > >
> > > > *** REFERENCES
> > > >
> > > > [1] Ren, J., Poulenard, A., Wonka, P., and Ovsjanikov, M. Continuous and orientation-preserving correspondences via functional maps. ACMTrans. Graph. 37, 6 (dec 2018)

---

### Official Review · Reviewer_Y9CP · 2022-07-11

**Rating:** 6
**Confidence:** 3
**Soundness:** 2 fair
**Presentation:** 3 good
**Contribution:** 2 fair

**Summary:**

The paper presents a graph matching method using 2D graph convolution. In articular, the proposed method is designed to achieve better accuracy for extract shape matching. The method can be applied to both noisy non-isometric data and nearly-isometric shapes. Evaluations show that the proposed method achieves state of the art performance.

**Questions:**

- How sensitive is 2D-GEM to the correspondence initialization?
- As mentioned, GRAMPA is normally used as initialization, how does 2D-GEM perform if initialized with GRAMPA, instead of using SHOT with GMWM?
- 2D-GEM could use point features, does this help on SCAPE or TOPKIDS? Intuitively, pointwise features are potentially more noisy than pairwise features?
- The ablation on non-isometric shapes in Appendix E.2 shows that 2-Hop algorithm is as good as 2D-GEM. I found this disturbing as this effectively invalid the contributions of 2D-GEM. Similarly, in near-isometric case, it seems that LMD is the most important module for the performance gain.

**Limitations:**

- Lack of theoretical support for good performance for exact shape matching.
- The proposed method borrows various techniques from previous methods in order to tackle both isometric and non-isometric case. It achieves the state of the art performance, but it is clear that these performance gain are not due to the 2D Convolution.


**Strengths And Weaknesses:**

Strengths:
- Shape matching in 3D has been well studied. The paper tackles the case for exact matching, in which previous methods don't perform well.
- The quality of the evaluation is good. Comparisons to competing methods and self ablations are extensive.
- The proposed algorithm include multiple steps involving various components, though the authors could well deliver the essential concepts. Notably, the relation between 2D-GEM and GRANMPA helps to understand the novelty.

Weaknesses:
- Lack of theoretic foundations, it is not clear why the proposed method achieve excellent performance for exact matching on the selected datasets.
- Comparing to related works, the proposed pipeline has high similarity with GRAMPA [18], in terms of the use of cost function, 2-hop improvement, etc. LMD [73] is borrowed to adapt to near-isometric case.

---

> ### Author Response · Authors · 2022-08-02
> **Responses to Reviewer Y9CP**
>
> **Question 1.** How sensitive is 2D-GEM to the correspondence initialization?
>
> **Response.** As shown in Appendix D, 2D-GEM is robust to initializations on graphs and on shapes.  When no features or initialization are present, 2D-GEM is shown to be comparable with the  spectral baseline GRAMPA [1] . However, when an informative initialization or features is present, 2D-GEM performs significantly  better. Moreover, we added Appendix F to the paper to show the effects of initialization on 2D-GEM. As shown by the  plots included in this Appendix F,  2D-GEM is  robust to initializations on 3D-shapes as well. Due to the the that open-review does not allow us to upload figures, we kindly refer the reviewer to the newly added Figures 9 and 10 in the paper.
>
> **Question 2.** As mentioned, GRAMPA is normally used as initialization, how does 2D-GEM perform if initialized with GRAMPA, instead of using SHOT with GMWM?
>
> **Response.**  2D-GEM achieves good performance as well when initialized with GRAMPA, although in this case we have to increase the iterations on isometric shapes to 60 instead of 10 as used in the paper. We refer the reviewer to the newly added Figures~9 and 10 in the paper.
>
> **Question 3.** 2D-GEM could use point features, does this help on SCAPE or TOPKIDS? Intuitively, pointwise features are potentially more noisy than pairwise features?
>
> **Response.** Though in the paper we use the initialization, we added Appendix F showing that the performance is comparable when using the SHOT features directly . We refer the reviewer to the newly added Figures 9 and 10 in the paper. One of the curves labelled Dshot in In these plots  is that of using point-wise features SHOT directly with the 2D-convolution. These show that 2D-GEM is robust to the noises in the initialization, since 2D-GEM retained a good performance in-spite of which initialization was used.
>
> **Question 4.** The ablation on non-isometric shapes in Appendix E.2 shows that 2-Hop algorithm is as good as 2D-GEM. I found this disturbing as this effectively invalid the contributions of 2D-GEM. Similarly, in near-isometric case, it seems that LMD is the most important module for the performance gain.
>
> **Response.** As shown in Appendix D, 2D-GEM by itself performs well compared with other baselines even when no informative features are present, and when informative features or initialization are present, it performs better. However, to boost the performance even further on shape matching in general, we employed the 2-Hop algorithm for non-isometric shapes as done by the strong baseline GRAMPA, and on isometric shapes we further constrained the framework with the LMD as done with the strong baseline DIR. However, note that (a) 2D-GEM performs better than GRAMPA and DIR[2]  even when using the same parameters as shown in Figure~5 in the paper, and (2) both the LMD and 2-Hop depend on how good the algorithm with which they are paired performs, since they are both only boosters.
>
> **Limitations 1.** Lack of theoretical support for good performance for exact shape matching. The proposed method borrows various techniques from previous methods in order to tackle both isometric and non-isometric case. It achieves the state of the art performance, but it is clear that these performance gain are not due to the 2D-Convolution.
>
> **Response.**  Indeed a weakness of our work as pointed out in our limitations is the fact that unlike GRAMPA and other baselines, we do not provide any solid theoretical exact recovery guarantees. However, to address the reviewer's  concern on the justification of 2D-GEM, we first recall that in Sections 2 and 3 of the paper we motivate 2D-GEM from the fact that  graph convolution  is naturally known to be a denoising operation, we reformulate noise in the context of graph matching. Second, we validate our concept in Appendix~D on random Erdos Regny graphs. Third, 2D-GEM  provides a simple theoretical framework to understand spectral baselines such as GRAMPA. Fourth, the 2-Hop algorithm is a booster that relies on innitializations, it greatly depends on the algorithm with which it is paired. Likewise the LMD is used to enforce a constraint and is not a map updating algorithm by itself. Fifth, we recall the fact that in Figure 5 of the paper 2D-GEM outperformed the Baselines GRAMPA and DIR which both used the 2Hop and the LMD respectively. We equally tried DIR, GRAMPA and 2D-GEM using the code provided and setting maxiter 60 on isometric and non-isometric datasets, as well as the same LMD thresholds for DIR and 2D-GEM, it is clear that 2D-GEM and the results are similar to Figure 5 in the paper. Lastly, we acknowledge the reviewer's concern once again, since future in-depth theoretical analysis on exact recovery guarantees for 2D-GEM will indeed be very valuable to the research community as it will serve as a bridge to provide recovery guarantees on graphs when features are available.
>
> Thank you so much again for all your comments.

---

> > ### Author Response · Authors · 2022-08-02
> > **Responses to Reviewer Y9CP**
> >
> > ***
> >
> > **References**
> >
> > [1] Fan, Z., Mao, C., Wu, Y., and Xu, J. Spectral graph matching and regularized quadratic relaxations: Algorithm and theory. In proceedings of the 37th International Conference on Machine Learning (13–18 Jul 2020), H. D. III and A. Singh, Eds., vol. 119 of Proceedings of Machine Learning Research, PMLR, pp. 2985–2995.
> >
> > [2] Xiang, R., Lai, R., and Zhao, H. A dual iterative refinement method for non-rigid shape matching. In Proceedings of the IEEE/CVF Conference on Computer Vision and Pattern Recognition (CVPR) (June 2021), pp. 15930–15939.

---

> > ### Comment · Reviewer_Y9CP · 2022-08-07
> > **comments on author response**
> >
> > - Q1: It is difficult to understand Fig.10 and therefore the justification to my question 1.
> > - Q2: thanks for evaluating the method with new initialization method,  it is good to see that similar performance could be obtained at the cost of more iterations.
> > - Q3: The labels in Fig.9 is not clear to me. What do they mean using '+' symbol?
> > - Q4: my concert is about the ablations using 3D shapes, while the authors refer to results using simulated graphs.
> >
> > In general, Fig.9 and Fig.10 are lacking sufficient descriptions so it is rather hard to get the values of them. In terms of the performance boosting using 2-hop and LMD, the authors did not reply directly.

---

> > > ### Author Response · Authors · 2022-08-09
> > > **Thanks for further comments to Reviewer Y9CP**
> > >
> > > Q.1.It is difficult to understand Fig.10 and therefore the justification to my question 1.
> > >
> > > R.1. We have adjusted Figures 9, 10 and 13 by rescaling it and zooming in. As seen on the plots, 2D-GEM is robust to initializations, though the final accuracy of 2D-GEM for a given number of iterations may be higher for initializations of somewhat better quality.
> > >
> > > Q.2.thanks for evaluating the method with the new initialization method, it is good to see that similar performance could be obtained at the cost of more iterations.
> > >
> > > R.2. Thank you for your comments and suggestions again.
> > >
> > > Q.3. The labels in Fig.9 are not clear to me. What do they mean using '+' symbol?
> > >
> > > R.3. We have adjusted the labels as suggested by the Reviewer. The '+' means using the features with the GMWM to initialize $P^{0}$ for 2D-GEM, while D means using the features to build $XaXb’$ and using the 2D-Conv as in equation 8 followed by the GMWM for the initialization.
> > >
> > > Q.4. My concern is about the ablations using 3D shapes, while the authors refer to results using simulated graphs.
> > > In general, Fig.9 and Fig.10 are lacking sufficient descriptions so it is rather hard to get the values of them.
> > >
> > > R.4. First, referring to the ablations on 3D shapes, the 2-Hop and LMS are both boosters used to enforce constraints on the 2D-GEM pipeline (2-Hop for neighborhood preservation and LMD for distortion preservation). In the ablation studies in Appendix E, we aim at showing why the framework as a whole is necessary, while in Appendix D we aim at validating the use of the 2D-Graph convolution and in Section 5 of the paper we aim at comparing 2D-GEM with strong baselines that use the same 2 constraints (GRAMPA with the 2-Hop, and DIR with the LMD). Second, referring to the previous question “2D-GEM could use point features, does this help on SCAPE or TOPKIDS? Intuitively, pointwise features are potentially more noisy than pairwise features?”, we are sorry for not understanding it properly at first. In effect, with the 2D-Convolution, 2D-GEM can use point-wise features. Fig.9 and Fig.10 show different point-wise descriptors (features) used to initialize $P^{0}$ for 2D-GEM, and it seems that on both Isometric (SCAPE) and Non-isometric (TOPKIDS) shapes, all descriptors achieve a reasonable performance with 2D-GEM.
> > >
> > > Q.5. In terms of the performance boosting using 2-hop and LMD, the authors did not reply directly.
> > >
> > > R.5. Should the Reviewer be referring to how much performance is gained by both methods, we would say on isometric shapes, the LMD seems to be a more essential constraint to our framework as setting smaller LMD threshold values (i.e., enforcing this constraint helps increase the performance to become higher than the 2-Hop in cases (2) and (3) of figure 8 in Appendix E, while on non isometric shapes the reverse is true (in the same cases of figure 7). However, we remark that 2-hop and LMD are only boosters (used to enforce constraints)  and cannot be used by themselves.
> > >
> > > Thank you again for your comments and insights

---

### Official Review · Reviewer_aRMb · 2022-07-11

**Rating:** 6
**Confidence:** 4
**Soundness:** 3 good
**Presentation:** 3 good
**Contribution:** 3 good

**Summary:**

The paper proposes a shape-correspondence approach based on combined  spectral-domain filtering of the Laplace-Beltrami operators and the feature assignment in the corresponding spectral embedding space. The ideas per se are not new, but what I believe made a big difference is the adoption of the joint spectral filter $\omega(\lambda_i,\mu_j,e)$ that kills the signal associated to corresponding eigenvalues that are different from one another, thus not using noisy eigenfunctions in the spectral embedding.

**Questions:**

It might be interesting to learn the joint spectral filter.

**Limitations:**

Negative societal impacts have not been mentioned, but apart from the use in a futuristic 3D surveillance system, I cannot see one myself.
The authors are honest about the extraction of non-smooth solutions, but I think the phenomenon warrants a more in-depth analysis/explanation.

**Strengths And Weaknesses:**

Very strong experimental results
very nice idea of adopting the joint spectral filter which I really believe is the key to the performance.

The paper sees to want to reinvent spectral filtering on graphs and linear assignment on the spectral embedding space with slightly different names.

---

> ### Author Response · Authors · 2022-08-02
> **Responses to Reviewer aRMb**
>
> **Questions 1.** It might be interesting to learn the joint spectral filter.
>
> **Response** Thank you for your suggestion. Considering the connection between 2D-GEM and classical existing Graph Neural Networks (GNNs), this would be an interesting idea to explore for future work. This gives us the opportunity to study that just like classical GNNs, the 2D-spectral graph filter in 2D-GEM may be used for a variety of correspondence tasks associated with graphs with or without features.
>
> **Limitations 1.** Negative societal impacts have not been mentioned, but apart from the use in a futuristic 3D surveillance system, I cannot see one myself. The authors are honest about the extraction of non-smooth solutions, but I think the phenomenon warrants a more in-depth analysis/explanation.
>
> **Response** Concerning the non-smooth final maps produced by the algorithm, we will  try to dive deeper into that in future work. However, for now, we note that the problem may arise due to the initial map being very poor and non smooth as pointed out by Consistent ZoomOut [1]. In most cases, this may be reduced to an extent by (a) either using 2D-GEM in a ZoomOut-like manner, i.e., starting with lower eigen-values and iteratively increasing them (though this strategy works in some cases, it may not always work as pointed out in [1]), or by (b) using more rounds of our pipeline, i.e., the LMD (since the LMD is an effective measure of the continuity and distortion of the map), the map update via convolution (for denoising the map),  and the 2-Hop algorithm (since the 2-Hop algorithm also enforces neighbourhood preservation).
> Due to the fact that open-review does not allow us to upload figures, we kindly refer the reviewer to the newly added Figures 11 and 12 in the paper.
>
> ***
> **References**
>
> [1] Huang, R., Ren, J., Wonka, P., and Ovsjanikov, M. Consistent zoomout: Efficient spectral map synchronization. In Proc. SGP (2020),
> vol. 39.

---

> > ### Comment · Reviewer_aRMb · 2022-08-08
> > **Comments on Authors' response**
> >
> > I want to thank the authors for their detailed answers. indeed I am looking forward to the discussed development in their future work

---

> > > ### Author Response · Authors · 2022-08-09
> > > **Thanks for further comments to Reviewer aRMb**
> > >
> > > Thank you again for your comments and suggestions.

---

### Author Response · Authors · 2022-08-02
**We have uploaded a revised version [Updated on 03 Aug.]**

Dear reviewers,


We have revised our paper following the suggestions/comments from all the reviewers. These changes are highlighted in blue in the updated paper. Some changes are:

- A discussion of  Fast sinkhorn filters [1] in Appendix A.2.
- A discussion of Structured regularization of functional map computations [2] in Appendix A.2.
- A study on the effects of different LMD [3] thresholds on 2D-GEM in Appendix H.
- A study on the effects of different initializations on 2D-GEM as well as a brief comment on re-meshed shapes and shapes with different connectivity in Appendix F.
- A study on the effects of 2D-GEM on an initial non-smooth map in Appendix G.
- We also fixed some typos that will not affect the other findings and results.

***
**References**

[1] Pai, G., Ren, J., Melzi, S., Wonka, P., and Ovsjanikov, M. Fast sinkhorn filters: Using matrix scaling for non-rigid shape correspondence
with functional maps. In Proceedings of the IEEE/CVF Conference on Computer Vision and Pattern Recognition (CVPR) (June 2021), pp. 384–393.

[2] Ren, J., Panine, M., Wonka, P., and Ovsjanikov, M. Structured regularization of functional map computations. Comput. Graph. Forum 38, 5 (2019), 39–53.

[3] Xiang, R., Lai, R., and Zhao, H. A dual iterative refinement method for non-rigid shape matching. In Proceedings of the IEEE/CVF Conference on Computer Vision and Pattern Recognition (CVPR) (June 2021), pp. 15930–15939.

---

> ### Author Response · Authors · 2022-08-09
> **We have uploaded a revised version [Updated on 08 Aug.]**
>
> Dear reviewers,
>
>
> We have revised our paper following the suggestions/comments from all the reviewers. These changes are highlighted in red in the updated paper. Some changes are:
>
> - Citations of more related work in Section 1 including [1] and [2].
> - Statement on novelty of the work in Section 2.
> - Comments on different initializations in Section 4.1.
> - More Samples used for plots in figure 9.
> - We also fixed some typos and references that will not affect the other findings and results.
>
> ***
> **References**
>
> [1] Pai, G., Ren, J., Melzi, S., Wonka, P., and Ovsjanikov, M. Fast sinkhorn filters: Using matrix scaling for non-rigid shape correspondence
> with functional maps. In Proceedings of the IEEE/CVF Conference on Computer Vision and Pattern Recognition (CVPR) (June 2021), pp. 384–393.
>
> [2] Ren, J., Panine, M., Wonka, P., and Ovsjanikov, M. Structured regularization of functional map computations. Comput. Graph. Forum 38, 5 (2019), 39–53.

---

### Meta-Review · Area_Chair_dCV4 · 2022-08-26

**Recommendation:** Accept
**Confidence:** Certain

**Metareview:**

All the reviewers agreed that the paper is a nice addition to the shape matching literature. While its contributions are a bit incremental, they highlighted the quality of the evaluation, and the introduction of ML methods in the graphics community should stimulate follow-up works.

**Award:**

No

---

### Decision · Program_Chairs · 2022-09-14

Accept